# A Simplified Algorithm for Setting the Observer Parameters for Second-Order Systems with Persistent Disturbances Using a Robust Observer

**DOI:** 10.3390/s22186988

**Published:** 2022-09-15

**Authors:** Alejandro Rincón, Fredy E. Hoyos, John E. Candelo-Becerra

**Affiliations:** 1Grupo de Investigación en Desarrollos Tecnológicos y Ambientales—GIDTA, Facultad de Ingeniería y Arquitectura, Universidad Católica de Manizales, Carrera 23 No. 60-63, Manizales 170002, Colombia; 2Grupo de Investigación en Microbiología y Biotecnología Agroindustrial—GIMIBAG, Instituto de Investigación en Microbiología y Biotecnología Agroindustrial, Universidad Católica de Manizales, Carrera 23 No. 60-63, Manizales 170002, Colombia; 3Departamento de Energía Eléctrica y Automática, Facultad de Minas, Universidad Nacional de Colombia, Sede Medellín, Carrera 80 No. 65-223, Robledo, Medellín 050041, Colombia

**Keywords:** state estimation, robust observer, bioprocess monitoring, nonlinear systems

## Abstract

The properties of the convergence region of the estimation error of a robust observer for second-order systems are determined, and a new algorithm is proposed for setting the observer parameters, considering persistent but bounded disturbances in the two observation error dynamics. The main contributions over closely related studies of the stability of state observers are: (i) the width of the convergence region of the observer error for the unknown state is expressed in terms of the interaction between the observer parameters and the disturbance terms of the observer error dynamics; (ii) it was found that this width has a minimum point and a vertical asymptote with respect to one of the observer parameters, and their coordinates were determined. In addition, the main advantages of the proposed algorithm over closely related algorithms are: (i) the definition of observer parameters is significantly simpler, as the fulfillment of Riccati equation conditions, solution of LMI constraints, and fulfillment of eigenvalue conditions are not required; (ii) unknown bounded terms are considered in the dynamics of the observer error for the known state. Finally, the algorithm is applied to a model of microalgae culture in a photobioreactor for the estimation of biomass growth rate and substrate uptake rate based on known concentrations of biomass and substrate.

## 1. Introduction

In the monitoring and control of biological and biochemical processes, it is crucial to have real-time knowledge of variables such as the concentrations of biomass, products or reactants; the growth rate of microorganisms; and the substrate consumption rate [1,2,3,4]. Online knowledge of the substrate uptake rate is needed for the application of automatic control [4], whereas online knowledge of the specific growth rate (μ) is usually required in the following cases: (i) in automatic control with the biomass concentration as the output [5]; (ii) in automatic control with μ as the output (see [6,7]); (iii) in the maximization of growth rate via an extremum seeking controller [8]; (iv) in the maximization of the gaseous outflow rate via an extremum seeking controller [9]. The concentrations and reaction rates can be estimated by using state observers combined with the measurement of some state variables, and a known mass balance model for the measured states [1,10,11,12]. Control design for multi-agent systems is another active area of observer design. In these systems, observers are designed to estimate the unmeasured states of adjacent agents [13,14,15]. 

Robust observer designs have been developed for tackling uncertainty in the dynamics of observation errors. Common designs consider: (i) a lack of knowledge in the dynamics of the observation error for the unknown state; (ii) an unknown term of the unknown state in the dynamics of the observation error for the known state [10,16,17,18,19,20,21,22]. In the case that the dynamics of the observation error for the known state involves an additional additive uncertain term and it is persistent but bounded, the steady state estimation involves error, even if a discontinuous observer is used. However, the estimation error depends on the observer parameters, so it can be reduced to a certain extent, provided that the known limits of the uncertainty [23,24]. 

In [25], a tuning procedure is proposed for setting the parameters of an extended state observer (ESO) for a closed loop second-order system with measurement noise and bounded external disturbance. The used observer is a Luenberger-like extended state, which is intended to estimate the external disturbance. The stability of the resulting estimation error dynamics is determined based on the state matrix, and two choices are proposed for setting the observer gains that lead to the real negative eigenvalues of the state matrix. Furthermore, the time-dependent bound of the transient behavior of the observation errors is determined, which gives the exponential convergence rate and the width of the convergence region in terms of observer parameters and some disturbances of the dynamics of the observation errors. However, the time derivative of the external disturbance is required to be bounded, and the input gain is required to be locally Lipschitz continuous. 

In [21], an observer is designed to estimate the parameters of the tunneling current in a Scanning Tunneling Microscope (STM). The measurement of the tunneling current ($i_{t}$) is described by additive noise and first-order sensor dynamics, with the current $i_{t}$ as its input. The observer is used for estimating the tunneling current $i_{t}$. The estimate is expressed in terms of Laplace transforms of the actual current and additive noise. The observer parameters must be chosen to have poles with strictly negative real parts in the transfer functions. A simple choice is proposed, which yields a second-order characteristic polynomial with a damping coefficient of one. However, no external disturbance is considered in the first-order sensor dynamics. 

In [10], a filtered high gain observer is designed for a class of non-uniformly observable systems, and then it is applied to a phytoplanktonic growth model. Bounded disturbances and noisy measurements are considered, although bounded disturbances are not considered in the dynamics of the known state. The observation errors exponentially converge to a compact set whose width is a function of the observer parameters and bounds of the disturbances and measurement noise. Thus, the boundary of the transient behavior of the observation errors is determined. Furthermore, an improved observer is formulated for the case of sampled outputs, and the transient boundary of the observation errors is determined. Finally, the observer is applied to a model of continuous culture of phytoplankton, with an estimation of the substrate and cell quota concentrations based on biomass concentration measurements. The main limitations of the observer design are: (i) bounded disturbances are not considered in the dynamics of the observation error for the known state; (ii) some conditions of a Riccati differential equation must be satisfied.

In [23], a super-twisting observer is designed for a two-dimensional system, considering disturbances in the two observation error dynamics. Two disturbance types are considered in the observation error dynamics: in the first, the upper bound is the function of the observation error for the known state (known observation error); in the second, the upper bound is constant. The observation errors converge to zero in finite time for the first disturbance type. Thus, the convergence is proved, and the convergence time is determined. The observation errors converge to a compact set for the second disturbance type. Thus, the convergence region is determined, but the time-dependent bound of the transient behavior of the observation errors and the convergence times are not determined. In addition, the observer design algorithm is proposed, which involves the selection of design parameters. Finally, the observer and the algorithm are applied to a model of microalgae culture in a photobioreactor, performing an estimation of biomass growth rate, substrate uptake rate, and internal quota. However, the observer algorithm involves an iterative solution of LMIs, and the observer involves a discontinuous signal. 

In this work, a new algorithm is proposed for setting the parameters of a robust observer for second-order systems, considering persistent but bounded disturbances in the two observation error dynamics. The algorithm is applied to a model of microalgae culture in a photobioreactor for the estimation of biomass growth rate and substrate uptake rate based on known concentrations of biomass and substrate. The main contributions over closely related studies of the stability of state observers (for instance [10,21,23,25]) are: Ci. The width of the convergence region of the observer error for the unknown state is expressed in terms of the interaction between the observer parameters and the disturbance in terms of the observer error dynamics. Thus, the desired estimation accuracy can be defined by the user by setting the observer parameters in accordance with this relationship. In contrast:
−In [21], the dependence of the width of the convergence region on the observer parameters and disturbance terms is not determined.−In [10], the width of the convergence region is expressed in terms of the bounds of the disturbances and measurement noise, but bounded disturbances are not considered in the dynamics of the estimation error of the known state, so the effect of these disturbances is not considered in the convergence region. −In [23], the width of the convergence region is expressed in terms of the bounds of the disturbances of the dynamics of the estimation error of the known state, but the effect of observer parameters and bounded disturbance of the dynamics of the estimation error of the unknown state are not considered.Cii. The properties and limits of this width are determined. It was found that this width has a minimum point and a vertical asymptote with respect to one of the observer parameters, and their coordinates were determined. Then, the highest accuracy of the state estimation can be obtained by setting the observer parameters equal or similar to the coordinates of the minimum. In contrast, in [21,23,25]: (i) the properties, limits, and minimum of the width of the convergence region of the estimation error for the unknown state are not determined; (ii) the observer parameter values that lead to the lowest width of the convergence region are not determined.Ciii. The algorithm considers the combined effect of disturbance terms and observer parameters on the width of the convergence region.

In addition, the advantages of the proposed algorithm over closely related algorithms are: (i)***Advantage A1***. It involves a significantly simpler definition of observer parameters: the fulfillment of Riccati equation conditions, solution of LMI constraints, and fulfillment of eigenvalue conditions are not required, thus, reducing the trial-and-error effort. In contrast, the fulfillment of these conditions is commonly required in closely related observer strategies, for instance [10,23,26].(ii)***Advantage A2***. Different from [10,16,17,18,19,20,21,22], unknown bounded terms are considered in the dynamics of the observer error for the known state.(iii)***Advantage A3***. The time derivatives of the disturbance terms of the plant model are not required to be bounded, whereas this condition is required in [25,27,28,29,30].(iv)***Advantage A4***. Different from [23,31], discontinuous signals are not used in the observer, thus, avoiding problematic numerical solutions.

The work is organized as follows. The bioreactor model is presented in Section 2. The preliminaries in Section 3 include the observer equations and the bound of the transient and asymptotic behavior of the observer error. The main results presented in Section 4 include the formulation of the algorithm for setting the observer parameters and the determination of the width of the convergence region of the observer error in terms of the observer parameters. An application to a microalgae bioreactor is shown in Section 5, and the discussion and conclusions are drawn in Section 6.

## 2. Bioreactor Model

Consider the system
(1)$$\frac{dx_{1}}{dt}=h_{1}+bx_{2}+\delta_{1},$$
(2)$$\frac{dx_{2}}{dt}=h_{2}+\delta_{2},$$
where $x_{1}$ and $x_{2}$ are the states; $h_{1}$ and $h_{2}$ are model functions; $\delta_{1}$ and $\delta_{2}$ are disturbance terms; and b is the $x_{2}$ gain in the dynamics of $x_{1}$. The model terms fulfill the following assumptions: 

**Assumption** **1.**
*The functions*

$h_{1}$

*and*

$h_{2}$

*are known; the state*

$x_{1}$

*is measured, and the coefficient*

b

*is known; the state*

$x_{2}$

*and the terms*

$\delta_{1}$

*and*

$\delta_{2}$

*are unknown.*


**Assumption** **2.**
*The coefficient b is bounded away from zero:*

$$\left|b\right|\geq b_{min}>0\;$$

*where*

$b_{min}$

*is an unknown positive constant.*


**Assumption** **3.**
*The disturbance terms*

$\delta_{1}$

*and*

$\;\delta_{2}$

*are bounded.*


## 3. Preliminaries: Observer and Bounds for the Transient and Steady Behavior of the Observer Error

In this section, the observer equations and the bounds for the transient and asymptotic behavior of the observer error for the unknown state (${\bar{x}}_{2}$) are recalled from [24]. The detailed mathematical procedure of the Lyapunov-based formulation of the observer is provided in [24].

### 3.1. Observer

The observer equations are [24]:(3)$$\frac{d{\hat{x}}_{1}}{dt}=b{\hat{x}}_{2}-\left|b\right|\left(\omega {\bar{x}}_{1}+\left(k+\frac{1}{4\omega }\right)\psi_{x_{1}}+sat_{x_{1}}\hat{\theta }\right)+h_{1},$$
(4)$$\frac{d{\hat{x}}_{2}}{dt}=-b\omega \left(\left(k+\frac{1}{4\omega }\right)\psi_{x_{1}}+sat_{x_{1}}\hat{\theta }\right)+h_{2},$$
(5)$$\frac{d\hat{\theta }}{dt}=\gamma \left|b\right|\left|\psi_{x_{1}}\right|.$$
where
(6)$${\bar{x}}_{1}={\hat{x}}_{1}-x_{1},$$
(7)$$\psi_{x_{1}}=\{\begin{array}{c} {\bar{x}}_{1}-\varepsilon \;for\;{\bar{x}}_{1}\geq \varepsilon \;\\ 0\;for\;{\bar{x}}_{1}\;\epsilon \;\left[-\varepsilon ,\;\varepsilon \right]\;\\ {\bar{x}}_{1}+\varepsilon \;for\;{\bar{x}}_{1}\leq -\varepsilon \end{array}$$
(8)$$sat_{x_{1}}=\{\begin{array}{c} 1\;for\;{\bar{x}}_{1}\geq \varepsilon \;\\ \frac{1}{\varepsilon }{\bar{x}}_{1}\;for\;{\bar{x}}_{1}\;\in \\ -1\;for\;{\bar{x}}_{1}\leq -\varepsilon \; \end{array}\left[-\varepsilon ,\;\varepsilon \right]$$
σ=sign(b)
where ${\hat{x}}_{1}$ is the estimate of $x_{1}$, ${\hat{x}}_{2}$ is the estimate of $x_{2}$, $\hat{\theta }$ is the updated parameter, and: (i) γ, k, ω are user-defined positive constants; (ii) the width of the convergence region of ${\bar{x}}_{1}$, that is, ε, is user-defined, positive, and constant. 

### 3.2. Mathematical Definitions

The main mathematical definitions are given as follows. b, $h_{1}$, $h_{2}$, $\delta_{1}$ and $\delta_{2}$ are terms of the model (1) and (2) described after Equations (1) and (2), satisfying Assumptions 1, 2, and 3. In addition, ${\bar{x}}_{1}={\hat{x}}_{1}-x_{1}\;$ is the observer error for the known state, ${\bar{x}}_{2}={\hat{x}}_{2}-x_{2}\;$ is the observer error for the unknown state; $x_{1}$ is the known state, $x_{2}$ is the unknown state, and $z={\bar{x}}_{2}-\sigma \omega {\bar{x}}_{1}$.

Mathematical definitions related to the function $V_{z}$:$$V_{z}=\frac{1}{2}\psi_{z}^{2}$$
$$\psi_{z}=\{\begin{array}{c} z+\delta_{min}\;for\;z\geq -\delta_{min}\geq 0\;\\ 0\;for\;z\;\in \left(-\delta_{max},-\delta_{min}\right)\\ z+\delta_{max}\;for\;z\leq -\delta_{max}\leq 0\; \end{array}$$
where $\delta_{min}\;$and $\delta_{max}$ are constants that satisfy
$$\delta \geq \delta_{min},\;\;\;\;\delta_{min}\in (-\infty ,0];\delta \leq \delta_{max},\;\;\;\;\delta_{max}\in [0,\;\;\infty ),$$
where δ is defined as
$$\delta =\frac{1}{b}\left(\frac{\delta_{2}}{\sigma \omega }-\delta_{1}\right),$$
and $\psi_{zo}$ is the value of $\psi_{z}$ at the initial time.

Mathematical definitions related to the overall Lyapunov function:$$V_{z\theta x1}=V_{z}+V_{x1}+V_{\theta }\;,$$
$$V_{x1}=\frac{1}{2}\psi_{x1}^{2};\;V_{\theta }=\frac{1}{2}\gamma^{-1}{\tilde{\theta }}^{2};\;\tilde{\theta }=\hat{\theta }-\theta$$
where θ is a positive constant fulfilling:$$\left|-\delta_{zt}-\delta_{1}/b\right|\leq \theta ;\;\delta_{zt}=\psi_{z}-z.$$

Mathematical definitions of convergence regions: $$\Omega_{x1}=\left\{{\bar{x}}_{1}:-\varepsilon \leq {\bar{x}}_{1}\leq \varepsilon \right\}$$
$$\Omega_{x2}=\left\{{\bar{x}}_{2}:\left|{\bar{x}}_{2}\right|\leq max\left\{-\delta_{min},\delta_{max}\right\}+\omega \varepsilon \right\}$$

### 3.3. Convergence of the Observer Error for the Known State

The combined state z is defined as:(9)$$z={\bar{x}}_{2}-\sigma \omega {\bar{x}}_{1},$$

The Lyapunov function for z is defined as
(10)$$V_{z}=\frac{1}{2}\psi_{z}^{2},$$
$$\psi_{z}=\{\begin{array}{c} z+\delta_{min}\;for\;z\geq -\delta_{min}\geq 0\;\\ 0\;for\;z\;\in \left(-\delta_{max},-\delta_{min}\right)\\ z+\delta_{max}\;for\;z\leq -\delta_{max}\leq 0\; \end{array}\;.$$
where $\delta_{min}\;$and $\delta_{max}$ are constant limits for the disturbance term
$$\delta =\frac{1}{b}\left(\frac{\delta_{2}}{\sigma \omega }-\delta_{1}\right),\;$$
that satisfy
$$\delta \geq \delta_{min},\;\;\;\;\delta_{min}\in (-\infty ,0],\;\delta \leq \delta_{max},\;\;\;\;\delta_{max}\in [0,\;\;\infty ).$$

Differentiating $V_{z}$ with respect to time, yields
$$\frac{dV_{z}}{dt}\leq -2\omega \left|b\right|V_{z}\leq 0.\;$$

The overall Lyapunov function is:$$V_{z\theta x1}=V_{z}+V_{x1}+V_{\theta }\;,$$
$$V_{x1}=\frac{1}{2}\psi_{x1}^{2};\;V_{\theta }=\frac{1}{2}\gamma^{-1}{\tilde{\theta }}^{2};\;\tilde{\theta }=\hat{\theta }-\theta$$
where $V_{z}$ is given by Equation (10), γ is a user-defined positive constant, $\hat{\theta }$ is provided by Equation (5) and θ is a positive constant fulfilling$$\left|-\delta_{zt}-\delta_{1}/b\right|\leq \theta ;\;\delta_{zt}=\psi_{z}-z.$$

The time derivative of $V_{z\theta x1}$ leads to
(11)$$\frac{dV_{z\theta x1}}{dt}=\frac{d}{dt}\left(V_{z}+V_{x1}+V_{\theta }\right)\leq -kb_{min}\psi_{x1}^{2}\leq 0.$$

This indicates the asymptotic convergence of the observer error ${\bar{x}}_{1}$ to the compact set $\Omega_{x1}=\left\{{\bar{x}}_{1}:-\varepsilon \leq {\bar{x}}_{1}\leq \varepsilon \right\}$.

**Remark 1.***The*$\psi_{z}$*definition given after Equation (10) indicates that*$\psi_{z}$*,*$\psi_{z}^{2}$*and*$d\psi_{z}^{2}/dz$*exist and are continuous. The*$\psi_{x_{1}}$*definition (7) indicates that*$\psi_{x_{1}}$*,*$\psi_{x1}^{2}$*, and*$d\psi_{x1}^{2}/d{\bar{x}}_{1}$*exist and are continuous. Consequently,*$V_{z}$*,* $V_{x1}$*and the overall Lyapunov function*$V_{z\theta x1}$*exist and are continuous. A detailed determination of*$dV_{z\theta x1}/dt$*,*$dV_{z}/dt$*,*$dV_{x1}/dt$*,*$dV_{\theta }/dt$*is given in* [24].

**Remark 2**. *The term ‘overall Lyapunov function’ is used for the Lyapunov function that results from the addition of several quadratic or positive forms and whose time derivative indicates the convergence result of some state. This term is also used in* [32,33]*. Notice that this condition is only fulfilled by*
$V_{z\theta x1}$*, as follows from Equation (11).*

### 3.4. Bounds for the Transient and Steady Behavior of the Observer Error for the Unknown State

From the definition of z (9), it follows that the observer error ${\bar{x}}_{2}$ can be rewritten in terms of z and ${\bar{x}}_{1}$: $${\bar{x}}_{2}=z+\sigma \omega {\bar{x}}_{1}.$$

This leads to
$$\left|{\bar{x}}_{2}\right|\leq \left|z\right|+\omega \left|{\bar{x}}_{1}\right|.$$

Combining the dynamics of z and ${\bar{x}}_{1}$, yields
(12)$$\left|{\bar{x}}_{2}\right|\leq \left|\psi_{zo}\right|e^{-\omega b_{min}\left(t-t_{o}\right)}+max\left\{-\delta_{min},\delta_{max}\right\}+\omega \left|{\bar{x}}_{1}\right|\;$$
where $\psi_{zo}$ is the value of $\psi_{z}$ at the initial time, and $\delta_{min}\;$and $\delta_{max}$ are constant limits for the disturbance term
(13)$$\delta =\frac{1}{b}\left(\frac{\delta_{2}}{\sigma \omega }-\delta_{1}\right),\;$$
that satisfy
(14)$$\delta \geq \delta_{min},\;\;\;\;\delta_{min}\in (-\infty ,0],\;\delta \leq \delta_{max},\;\;\;\;\delta_{max}\in [0,\;\;\infty ).$$

$b_{min}$ is a constant limit for b that satisfies Assumption 2. 

Despite the fact that the convergence of ${\bar{x}}_{1}$ to $\Omega_{x1}=\left\{{\bar{x}}_{1}:-\varepsilon \leq {\bar{x}}_{1}\leq \varepsilon \right\}$ is asymptotic, one can consider that ${\bar{x}}_{1}\in \Omega_{x1}$ for some $t\geq T_{1}$, that is, $\left|{\bar{x}}_{1}\right|\leq \varepsilon$. Combining with Equation (12) yields the time-dependent bound for the transient behavior of the observer error ${\bar{x}}_{2}$:(15)$$\left|{\bar{x}}_{2}\right|\leq \left|\psi_{zo}\right|e^{-\omega b_{min}\left(t-t_{o}\right)}+max\left\{-\delta_{min},\delta_{max}\right\}+\omega \varepsilon \;for\;t\geq T_{1}\;$$

Hence, ${\bar{x}}_{2}$ converges asymptotically to the compact set
(16)$$\Omega_{x2}=\left\{{\bar{x}}_{2}:\left|{\bar{x}}_{2}\right|\leq max\left\{-\delta_{min},\delta_{max}\right\}+\omega \varepsilon \right\}\;$$
so that the limits of the convergence region $\Omega_{x2}$ are $max\left\{-\delta_{min},\delta_{max}\right\}+\omega \varepsilon$ and $-max\left\{-\delta_{min},\delta_{max}\right\}-\omega \varepsilon$. 

## 4. Formulation of the Algorithm for Setting the Observer Parameters and Determination of the Width of the Convergence Region of the Observer Error in Terms of the Observer Parameters

In this section: (i) the width of the convergence region $\Omega_{x2}$ (16) is expressed in terms of the interaction between the parameters of the observer (3)–(5) and $\delta_{1},\delta_{2}$, the disturbance terms of the observer error dynamics; (ii) an algorithm is formulated for setting the observer parameters ω, ε, γ, k. 

### 4.1. Determination of the Width of the Convergence Region of the Observer Error

From the definition of δ in Equation (13), it follows that
(17)$$\left|\delta \right|\leq \frac{1}{\omega }d_{2}+d_{1}\;$$
where $d_{1}$ and $d_{2}$ are positive constants that satisfy
(18)$$\left|\frac{\delta_{2}}{b}\right|\leq d_{2};\;\left|\frac{\delta_{1}}{b}\right|\leq d_{1}\;$$

From Equation (17) and the conditions on $\delta_{min}$ and $\delta_{max}$ (14), it follows that the $\delta_{min}$ and $\delta_{max}$ values can be chosen to be:$$\delta_{max}=\frac{1}{\omega }d_{2}+d_{1};\;\delta_{min}=-\delta_{max}$$
so that the terms in Equation (14) are fulfilled. Then, the convergence set $\Omega_{x2}\;$(16) becomes
(19)$$\Omega_{x2}=\left\{{\bar{x}}_{2}:\left|{\bar{x}}_{2}\right|\leq f_{w}\;\right\}\;$$
(20)$$f_{w}=\frac{1}{\omega }d_{2}+d_{1}+\omega \varepsilon$$
so that the width of the convergence set $\Omega_{x2}$ is $f_{w}$ and the limits of $\Omega_{x2}$ are $-f_{w}$ and $+f_{w}$. The main features of the $f_{w}$ function are:(21)$$f_{w}>0;\;f_{w}has\;a\;vertical\;asymptote,\;at\;\omega =0\;\underset{\omega \to \infty }{\lim}f_{w}=\infty ;\;f_{w}>d_{1}\geq \underset{t\geq t_{o}}{\sup}\;\left|\delta_{1}/b\right|;\;\underset{\omega \to 0^{+}}{\lim}f_{w}=\infty \;$$

From these properties, it follows that $f_{w}\;$ has a minimum point with respect to ω. Its coordinates are determined by differentiating $f_{w}$ expression (20) with respect to ω, which yields:(22)$$\omega^{*}=\sqrt{d_{2}}\;\frac{1}{\sqrt{\varepsilon }};f_{w}^{*}=2\sqrt{d_{2}}\sqrt{\varepsilon }+d_{1}\;$$

Therefore, the relationship between $f_{w}^{*}$ and $\omega^{*}$ is given by
(23)$$f_{w}^{*}=2d_{2}\frac{1}{\omega^{*}}+d_{1}\;$$

**Remark 3**. *The properties (21) of the*$f_{w}$*function and its minimum (22) indicate that *$\omega =\omega^{*}$*and a low *ε* value leads to low *$f_{w}$*, which implies a low width of the convergence region *$\Omega_{x2}$*, and consequently, a high quality of *$x_{2}$* estimation, as follows from Equations (19) and (20).*

**Remark 4**. *An overlarge*ω*value fulfilling*$\omega \gg \omega^{*}$*leads to: (i) fast convergence of the upper bound of*${\bar{x}}_{2}$*, as follows from Equation (15); (ii) a large*$f_{w}$*value, which implies a low quality of*${\bar{x}}_{2}$*estimation, as follows from Equations (19) and (20). Therefore, the choice of*ω*must take into account both the convergence rate and the width of the convergence region of*${\bar{x}}_{2}$. 

**Remark 5**. *A low*ε*value leads to low *$f_{w}^{*}$*, since *$f_{w}^{*}$* increases with respect to *ε*, as follows from Equation (22). However, overly small *ε* values lead to steeper slopes in the shape of the *$sat_{x_{1}}$* signal (8) of the observer, which implies that the numerical solution of the differential equation must use a smaller step size.*

**Remark 6**. *The*$f_{w}^{*}$*function (23) increases with *$d_{1}$*, whereas *$\omega^{*}$* is independent of *$d_{1}$*, as follows from Equation (22).*

In the case that $\delta_{1}=0$ in Equation (1), we have
$$\left|\delta \right|\leq \frac{1}{\omega }d_{2}$$
and one can use
$$\delta_{max}=\frac{1}{\omega }d_{2};\;\delta_{min}=-\delta_{max}$$

Then, Equation (19) becomes
(24)$$\Omega_{x2}=\left\{{\bar{x}}_{2}:\left|{\bar{x}}_{2}\right|\leq f_{w}\right\}\;$$
(25)$$f_{w}=\frac{1}{\omega }d_{2}+\omega \varepsilon$$

The resulting features of $f_{w}$ for $\delta_{1}=0$ are:$$f_{w}>0;\;f_{w}has\;a\;vertical\;asymptote,\;at\;\omega =0;$$
$$\underset{\omega \to \infty }{\lim}f_{w}=\infty ;\;\underset{\omega \to 0^{+}}{\lim}f_{w}=\infty$$

From these properties, it follows that $f_{w}$ has a minimum point with respect to ω. The coordinates of this minimum are determined by differentiating $f_{w}$ expression (22) with respect to ω, which yields:(26)$$\omega^{*}=\sqrt{d_{2}}\frac{1}{\sqrt{\varepsilon }};f_{w}^{*}=2\sqrt{d_{2}}\sqrt{\varepsilon }\;$$

### 4.2. Formulation of the Algorithm for Setting the Observer Parameters

The algorithm presented in Algorithm 1 allows us to set the observer parameters ω, ε, γ, and k, so as to define: (i) the convergence rate of ${\bar{x}}_{2}$; (ii) the value of $f_{w}=\frac{1}{\omega }d_{2}+d_{1}+\omega \varepsilon$, which is the width of the ${\bar{x}}_{2}$ convergence set $\Omega_{x2}=\left\{{\bar{x}}_{2}:\;\left|{\bar{x}}_{2}\right|\leq f_{w}\;\right\}$.
**Algorithm 1: **Algorithm for setting the parameters of the observer (3)–(5).
StepDescription1Cast the system model in the form (1)–(2) and identify the known state x1, the unknown state x2, and the terms b, h1, h2, δ1, δ2.2Obtain the values of bmin that satisfy Assumption 2 and the values of d2, d1, satisfying Equation (18). To this end, the values of d2, d1 can be obtained by the simulation of δ2/b, δ1/b, based on the x1, x2 model, with model parameter values obtained from either closely related studies or offline fitting.3Set the values of ω, ε to define:
−The time-dependent bound for the transient evolution of x¯2, given by Equation (15):|x¯2|≤|ψzo|e−ωbmin(t−to)+max{−δmin,δmax}+ωε; for t≥T1−The limit of the convergence region of x¯2, given by Equations (19) and (20):Ωx2={x¯2:|x¯2|≤fw }; fw=1ωd2+d1+ωε where fw>d1≥supt≥to |δ1/b| and the minimum point of fw is given by Equation (22):ω*=d21ε; fw*=2d2ε+d1
4Set a high value of γ to define the update rate of θ^, according to Equation (5). Set a high value of k to define the convergence rate of x¯1, according to Equation (11).

**Remark 7**. *The proposed algorithm and the observer (3)–(5) lead to a more practical and simpler real-time state estimation in either laboratory or industrial applications, according to the advantages A1 to A4, which are due to the observer of* [24]*. They can be applied to systems whose model can be cast in the second-order form (1) and (2), which includes a wide range of mechanical and physical systems. Some examples are:*−*Microalgae reactor represented by the Droop model: (i) estimation of specific bio-mass growth rate based on known biomass concentration; (ii) estimation of specific substrate uptake rate based on known substrate concentration—see [8,23]*.−*Anaerobic bioreactor for hydrogen production via the dark fermentation of glucose: estimation of influent glucose concentration based on known reactor glucose con-centration—see [3]*.−*Fed-batch bioreactor for ethanol production: (i) estimation of the rate of enzymatic hydrolysis based on known substrate (starch) concentration; (ii) estimation of the glucose consumption rate based on known glucose concentration—see [4]*.−*Membrane fuel cell: estimation of stack temperature based on known oxygen pres-sure—see [34]*.−*Photovoltaic system: estimation of the power gradient based on known generated electric power—see [35]*.−*DC-DC buck converter: estimation of the time derivative of the output tracking er-ror based on the known average output voltage—see [36]*.−*Second-order underactuated mechanical system: estimation of the time derivative of the pole angle—see [20]*.

**Remark 8**. *The convergence rate and the width of the*${\bar{x}}_{2}$*convergence set*$\left(f_{w}\right)$*can be properly defined by setting the observer parameters*ω*,*ε*,*γ*,*k*in accordance with the proposed procedure, with*ω*,*ε*values corresponding to the minimum point of*$f_{w}$*, that is,*$\omega^{*}$*,*$f_{w}^{*}$.

**Remark 9**. *The proposed observer algorithm deals with the combined effect of disturbance terms and observer parameters on the width of the convergence region for the estimation error of the unknown state, as follows*:(*a*)*The convergence region is expressed as a function of the combined effect of observer parameters and disturbance terms—see Equations (19) and (20). Then, the desired estimation accuracy can be defined by the user by properly setting the observer parameters in accordance with this relationship*.(*b*)*The properties of this expression are determined in terms of the observer parameters, including the limits and the coordinates of the minimum—see Equations (21)–(23). In turn, these properties allow choosing ω, ε values to avoid an undesired overlarge f_w value, and the lowest f_w value is obtained by using the coordinates of the minimum, that is, ω=ω^*, according to Equation (22). Thus, the highest accuracy of the state estimation is obtained by setting the observer parameters equal or similar to the coordinates of the minimum*.

## 5. Application to Microalgae Bioreactor

The developed algorithm for setting parameters of the observer (3)–(5) is used to estimate the substrate uptake rate ρ and specific growth rate μ in a continuous microalgae bioreactor. The concentrations of substrate and biomass are considered to be known, and the system is described by the Droop model [23,37]:(27)$$\frac{dx}{dt}=\mu x-Dx\;$$
(28)$$\frac{ds}{dt}=-\rho x+D\left(s_{i}-s\right)$$
(29)$$\frac{dq}{dt}=\rho -\mu q$$
where x is the biomass concentration, s is the substrate concentration, and q is the cell quota of assimilated nutrient; $D=F_{i}/v$ is the dilution rate, $F_{i}$ is the feeding flow rate, v is the broth volume, $s_{i}$ is the fed substrate concentration, μ is the specific growth rate, and ρ is the specific substrate uptake rate. The expressions for μ,ρ, and the model parameters are [23]:(30)$$\{\begin{array}{c} \mu \left(q\right)=\;\max\left\{0,\mu_{m}\left(1-\frac{Q_{O}}{q}\right)\right\};\rho =\rho_{m}\left(\frac{s}{s+K_{s}}\right)\\ \rho_{m}=0.03\frac{mgN}{mgC\cdot d};K_{s}=0.0010\frac{mgN}{L};\;\mu_{m}=0.5{\;d}^{-1};\\ Q_{O}=0.045\;\mathrm{mgN}/\mathrm{mgC};\\ x_{to}=0.1\;\mathrm{mgC}/L;s_{to}=0.01\;\mathrm{mgN}/L;\;q_{to}=0.06\;\mathrm{mgN}/\mathrm{mgC};\\ D=\{\begin{array}{c} 0.25\left(1+\sin\left(\frac{2\pi }{\tau_{D}}t\right)\right)\;\;\\ \;\;\;\;\;\;\;\;\;\;\;\;\;\;\;\;\;\;\;\;0\;for\;t\geq 6\;d \end{array}d^{-1}\;\mathrm{for}\;t<6\;d;\\ \tau_{D}=8\;d;s_{i}=0.05\;\mathrm{mgN}/L; \end{array}$$

The model details, including parameters and specific growth rate expression, are given in [23].

The $f_{w}$ curve as a function of ω and ε is computed using Equation (20), the curves of $f_{w}^{*}$ and $\omega^{*}$ as a function of ε are computed using Equation (22), and the $f_{w}^{*}\;\mathrm{vs}.\;\omega^{*}$ curve is computed using Equation (23).

The simulation of the model (27)–(29) and the observer (3)–(5) was performed using Matlab software (The Math Works Inc., Natick, MA, USA): the differential equations were numerically integrated using the ode45 routine. 

Although model (27)–(29) comprises three states, it leads to the following second-order subsystems:

A)$$\frac{ds}{dt}=-\rho x+D\left(s_{i}-s\right)$$
$$\frac{d\rho }{dt}=\delta_{2}$$
for the first example, so that $x_{1}=s;\;x_{2}=\rho$.

B)$$\frac{dx}{dt}=\mu x-Dx$$
$$\frac{d\mu }{dt}=\delta_{2}$$for the second example, so that $x_{1}=x;\;x_{2}=\mu$.

This approach is also considered in [23].

### 5.1. First Example: Estimation of Substrate Uptake Rate

The specific substrate uptake rate ρ is estimated using the substrate mass balance model (28) and the knowledge of substrate and biomass concentrations. The fed substrate concentration $s_{i}$ is inaccurately known: $s_{i}={\bar{s}}_{i}+\delta_{sin}$, where ${\bar{s}}_{i}$ is the known value of $s_{i}$, and $\delta_{sin}$ is the uncertainty; ${\bar{s}}_{i}=0.05$ mg N/L;$\;\delta_{sin}=0.1{\bar{s}}_{i}\times \sin\left(\frac{2\pi t}{\tau_{\mathrm{si}}}\right);\tau_{si}=3$. The substrate concentration (s) is the known state, and the specific substrate uptake rate (ρ) is the unknown state, so that substrate model (28) can be cast in the form (1), (2) with
(31)$$x_{1}=s;\;x_{2}=\rho ;\;b=-x;\;h_{1}=\left({\bar{s}}_{i}-s\right)D;\;h_{2}=0;\;\delta_{1}=D\delta_{si};\;\delta_{2}=\frac{d\rho }{dt}\;$$

Additionally, the observer (3)–(5) provides the estimate of ρ, that is, ${\hat{x}}_{2}=\hat{\rho }$. The observer structure is given in Figure 1. x is the biomass concentration, s is the substrate concentration, ρ is the specific substrate uptake rate, $D=F_{i}/v$ is the dilution rate, $F_{i}$ is the feeding flow rate, v is the broth volume, and $s_{i}$ is the fed substrate concentration. In addition, $x_{1}\;$is the known state, $x_{2}$ is the unknown state, and ${\hat{x}}_{2}$ is the estimate of $x_{2}$.

To examine the $f_{w}$ function (20), the $d_{1}$,$\;d_{2}$ bounds of the disturbance terms $\delta_{1}/b$, $\delta_{2}/b$ are obtained by simulation based on the $x_{1}$, $x_{2}$ model, yielding $d_{1}=0.04$, $d_{2}=0.11$. The curves of $f_{w}$, $f_{w}^{*}$, $\omega^{*}$ for different ε, ω values are shown in Figure 2. 

Low $f_{w}$ values are obtained for $\omega =\omega^{*}$ and low ε value (see Figure 2b,d), which is in accordance with remark 4.1. A minimum of the $f_{w}$ function is characterized by $\omega^{*}=8.564$, $f_{w}^{*}=0.0657$ for ε=0.0015, as follows from Equation (22). The observer parameters are chosen to be:(32)$$\varepsilon =0.0015;\omega =8.56;\;k=40;\gamma =100;\;{\hat{x}}_{1|to}=0.1;\;{\hat{x}}_{2|to}=0\;d^{-1};\;{\hat{\theta }}_{to}=0\;\;$$

So that $f_{w}\approx f_{w}^{*}$. The bioreactor is simulated using model (27)–(29) with plant model terms and parameters (30), whereas the observer (3)–(5) is simulated using the definition of the terms of the system model given by Equation (31), and the values of observer parameters given by Equation (32), and it is observed that (Figure 3):−The observer error ${\bar{x}}_{1}$ converges faster than ${\bar{x}}_{2}$.−The observer error ${\bar{x}}_{1}={\hat{x}}_{1}-x_{1}$ converges asymptotically to the compact set $\Omega_{x1}=\left\{{\bar{x}}_{1}:-\varepsilon \leq {\bar{x}}_{1}\leq \varepsilon \;\right\}$ and remains inside for t≥2.6 d approx. (Figure 3a,b). −The observer error ${\bar{x}}_{2}={\hat{x}}_{2}-x_{2}$ converges to the computed compact set $\Omega_{x2}$ and remains inside for t≥4.4 d approx. (Figure 3c,d). The limits $\left(-f_{w},+f_{w}\right)$ of the $\Omega_{x2}$ convergence set are indicated through dashed horizontal lines in Figure 3d. −The low width of $\Omega_{x2}$ is owed to the small values of $\delta_{1}/b$ and ε. 


The performed simulations confirm the adequacy of the parameter recommendations provided in the observer algorithm to achieve proper convergence speed and the width of the convergence region of ${\bar{x}}_{2}$. 

### 5.2. Second Example: Estimation of Biomass Growth Rate 

The specific growth rate μ is estimated using the biomass mass balance model (27) and the knowledge of biomass concentration. The biomass concentration (x) is the known state, and the specific growth rate μ is the unknown state, so the biomass model (27) can be cast in the form (1), (2) with
(33)$$x_{1}=x;\;x_{2}=\mu ;\;b=x;\;h_{1}=-Dx;\;h_{2}=0;\;\delta_{1}=0;\;\delta_{2}=\frac{d\mu }{dt}\;$$

Additionally, the observer (3)–(5) provides the estimate of μ, that is, ${\hat{x}}_{2}=\hat{\mu }$. The observer structure is given in Figure 4. x is the biomass concentration, s is the substrate concentration, μ is the specific growth rate, ρ is the specific substrate uptake rate, $D=F_{i}/v$ is the dilution rate, $F_{i}$ is the feeding flow rate, v is the broth volume, and $s_{i}$ is the fed substrate concentration. In addition, $x_{1}\;$is the known state, $x_{2}$ is the unknown state, and ${\hat{x}}_{2}$ is the estimate of $x_{2}$. 

To examine the $f_{w}$ function, the $d_{1}$,$\;d_{2}$ bounds of the disturbance terms $\delta_{1}/b$, $\delta_{2}/b$ are obtained by simulation based on the $x_{1}$, $x_{2}$ model, yielding $d_{1}=0$, $d_{2}=0.125$. The curves of $f_{w}$, $f_{w}^{*}$, $\omega^{*}$ for different ε, ω values are shown in Figure 5. 

Low $f_{w}$ values are obtained for $\omega =\omega^{*}$ and low ε value (see Figure 5b,d), which is in accordance with remark 4.1. A minimum of the $f_{w}$ function is characterized by $\omega^{*}=9.129$, $f_{w}^{*}=0.0097$ for ε=0.0015. The observer parameters are chosen to be:(34)$$\varepsilon =0.0015;\omega =9.12;\;k=40;\gamma =100;\;{\hat{x}}_{1|to}=0.1mgC/L;\;{\hat{x}}_{2|to}=0\;d^{-1};\;{\hat{\theta }}_{to}=0\;\;$$

So that $f_{w}\approx f_{w}^{*}$. The bioreactor is simulated using model (27)–(29) with plant model terms and parameters (30), whereas the observer (3)–(5) is simulated using the plant model terms and parameters given by Equation (33), and the values of observer parameters given by Equation (34), and it is observed that (Figure 6):−The observer error ${\bar{x}}_{1}$ converges faster than ${\bar{x}}_{2}$.−The observer error ${\bar{x}}_{1}={\hat{x}}_{1}-x_{1}$ enters to the compact set $\Omega_{x1}$ at 4.92 days and remains inside afterward (Figure 6a,b). −The observer error ${\bar{x}}_{2}={\hat{x}}_{2}-x_{2}$ remains inside its compact set for t≥15.3 d approx. (Figure 6c,d). The limits $\left(-f_{w},+f_{w}\right)$ of the $\Omega_{x2}$ convergence set are indicated through dashed horizontal lines in Figure 6d.−A low width of $\Omega_{x2}$ is achieved by choosing ε, ω values on the basis of the proposed algorithm.

## 6. Discussion and Conclusions

### 6.1. Discussion 

−The main contributions over closely related studies of the stability of state observers are: −Ci. The width of the convergence region of the observer error for the unknown state is expressed in terms of the interaction between the observer parameters and the disturbance in terms of the observer error dynamics. Then, the user defines the desired estimation accuracy by properly setting the observer parameters in accordance with the aforementioned relationship.−Cii. The properties and limits of this width are determined; it was found that this width has a minimum point and a vertical asymptote with respect to one of the observer parameters, and their coordinates were determined. Thus, the highest accuracy of the state estimation is obtained by setting the observer parameters equal or similar to the coordinates of the minimum.−The main challenges encountered in this research work are:−Choosing the idea of contributions Ci and Cii as the core topics of the paper required identifying contributions and limitations of high-quality works addressing observer design and stability analysis, mainly [23,25]. This implied a deep understanding of all the mathematical developments involved in the stability analysis, and also the advantages, disadvantages, and limitations of the observer and its stability properties.

The statement of the procedure for determining the constants that satisfy Equation (18). Based on several literature studies, we finally concluded that they could be obtained by the simulation of $\delta_{2}/b$,$\;\delta_{1}/b$, based on the $x_{1}$, $x_{2}$ model, with model parameter values obtained from either closely related studies or offline fitting, as stated in Algorithm 1.

### 6.2. Conclusions

In this work, a new algorithm is proposed for setting parameters of a robust observer for second-order systems, considering persistent but bounded disturbances in the two observation error dynamics. As the main contribution over closely related studies of the stability of state observers, the width of the convergence region of the observer error for the unknown state is expressed in terms of the interaction between the observer parameters and the disturbance terms of the observer error dynamics. Moreover, the properties and the minimum of this relationship were determined.

The proposed observer algorithm leads to a more practical and simpler state estimation in either laboratory or industrial applications. It can be used for systems whose model can be cast in second-order form, for instance, mechanical, chemical, and biochemical systems. Moreover, it can be used for multiple state estimation, in cases of several second-order systems. The choice of the observer parameters must consider both the convergence rate and width of the convergence region of the estimation error of the unknown state. Choosing the observer parameter values to be similar or equal to the values for the minimum leads to a high quality of state estimation. 

The performed simulations confirm the adequacy of the parameter recommendations provided in the observer algorithm to achieve the proper convergence speed and width of the convergence region of the observer error for the unknown state.

Future work will include: (i) extending the observer and the algorithm to system models of third order; (ii) extending the observer and the algorithm to system models of general nth order; (iii) considering noise in the measurement of the known state.

## Figures and Tables

**Figure 1 sensors-22-06988-f001:**
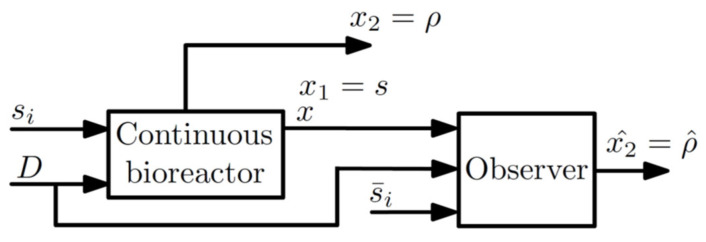
Structure of the observer application to microalgae bioreactor for estimating the specific substrate uptake rate ρ.

**Figure 2 sensors-22-06988-f002:**
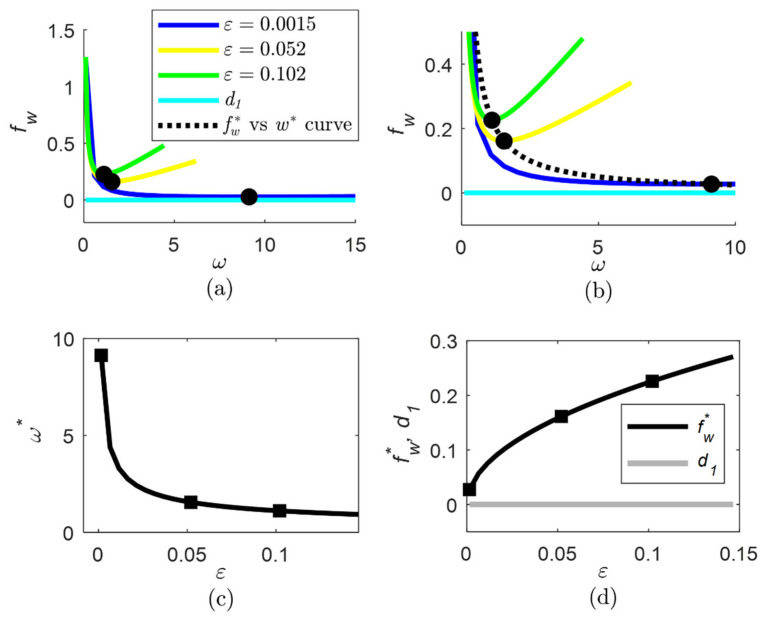
Effect of the observer parameters ε, ω on the $f_{w}$ function (20) for estimation of the specific substrate uptake rate ρ: (**a**) $f_{w}\;$ as a function of ω for several ε values, indicating the minimum point; (**b**) detail of $f_{w}\;$ as a function of ω for several ε values, indicating the minimum point; (**c**) values of $\omega^{*}$ as a function of ε, indicating the points for the ε values considered in subfigure a; (**d**) values of $f_{w}^{*}$ as a function of ε, indicating the points for the ε values considered in subfigure a.

**Figure 3 sensors-22-06988-f003:**
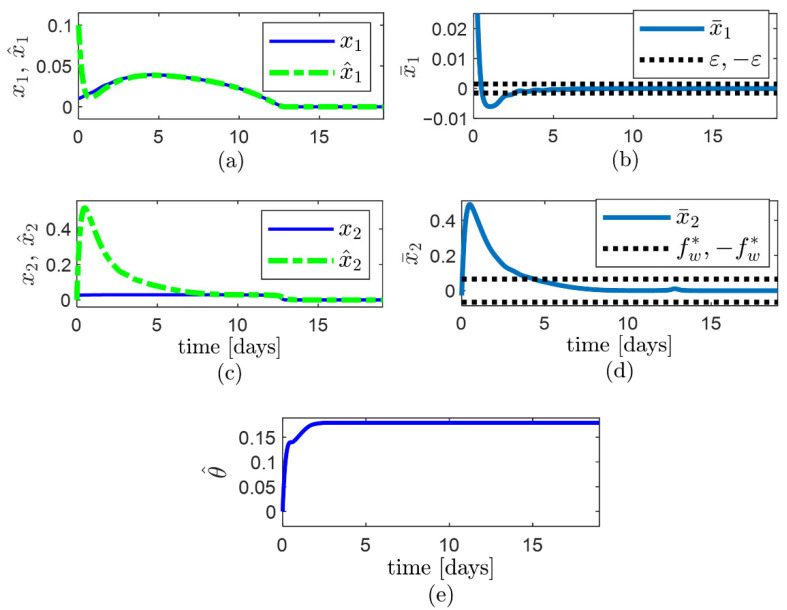
Performance of the observer (3)–(5) for estimation of specific substrate uptake rate ρ, using the observer parameters obtained through the proposed algorithm: (**a**) trajectory of state $x_{1}$ and estimate ${\hat{x}}_{1}$; (**b**) trajectory of the observer error for the known state, ${\bar{x}}_{1}$; (**c**) trajectory of state $x_{2}$ and estimate ${\hat{x}}_{2}$; (**d**) trajectory of the observer error for the unknown state, ${\bar{x}}_{2}$, with the limits $\left(-f_{w},+f_{w}\right)$ of the $\Omega_{x2}$ convergence set indicated through dashed horizontal lines; (**e**) trajectory of the updated parameter $\hat{\theta }$.

**Figure 4 sensors-22-06988-f004:**
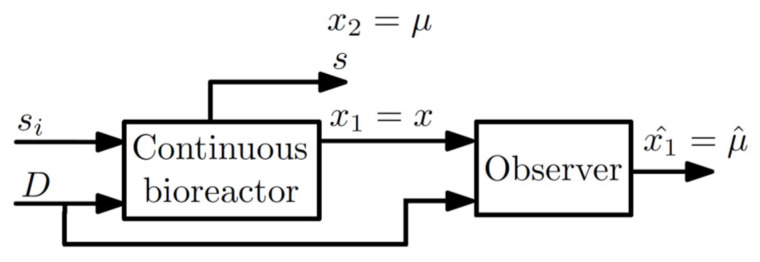
Structure of the observer application to microalgae bioreactor for estimation of the specific growth rate μ.

**Figure 5 sensors-22-06988-f005:**
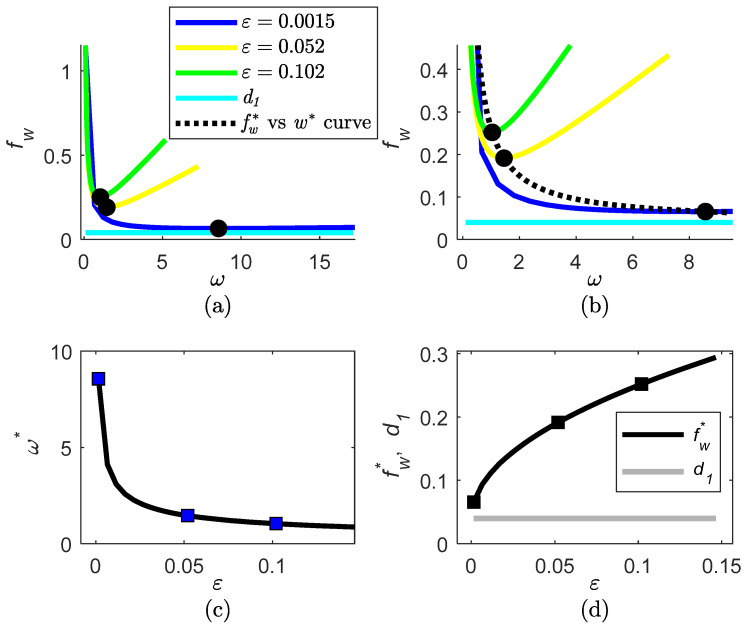
Effect of the observer parameters ε, ω on the $f_{w}$ function (19) for estimation of the specific growth rate μ: (**a**) $f_{w}\;$ as a function of ω for several ε values; (**b**) detail of $f_{w}\;$ as a function of ω for several ε values, indicating the minimum point; (**c**) values of $\omega^{*}$ as a function of ε, indicating the points for the ε values considered in subfigure a; (**d**) values of $f_{w}^{*}$ as a function of ε, indicating the points for the ε values considered in subfigure a.

**Figure 6 sensors-22-06988-f006:**
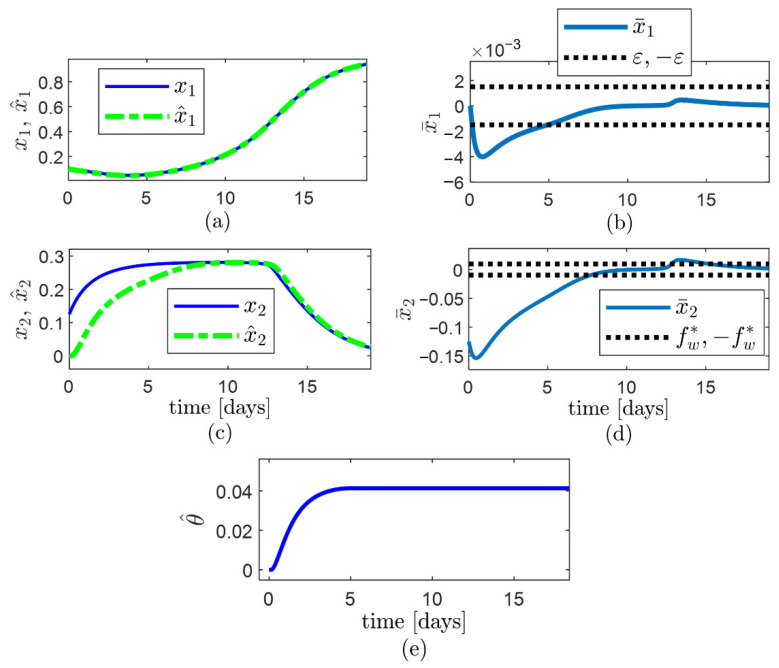
Performance of the observer (3)–(5) for estimation of specific growth rate μ, using the observer parameters obtained through the proposed algorithm: (**a**) trajectory of state $x_{1}$ and estimate ${\hat{x}}_{1}$; (**b**) trajectory of the observer error for the known state, ${\bar{x}}_{1}$; (**c**) trajectory of state $x_{2}$ and estimate ${\hat{x}}_{2}$; (**d**) trajectory of the observer error for the unknown state, ${\bar{x}}_{2}$, with the limits $\left(-f_{w},+f_{w}\right)$ of the $\Omega_{x2}$ convergence set indicated through dashed horizontal lines; (**e**) trajectory of the updated parameter $\hat{\theta }$.

## Data Availability

Not applicable.

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
