# Peer review of "A Simplified Algorithm for Setting the Observer Parameters for Second-Order Systems with Persistent Disturbances Using a Robust Observer"

_sensors, 2022, doi:10.3390/s22186988_

Round 1

Reviewer 1 Report

Comment on sensors-1888834

This manuscript proposes a new observer parameter setting algorithm to consider for persistent but bounded disturbances in the two observation error dynamics. The interaction between the observer’s parameters and disturbance terms of the observer error dynamics is used to represent the width of the observer error convergence region for the unknown state. The width has a minimum point and vertical asymptote with respect to an observer parameter, and their coordinates are determined. In addition, this algorithm takes into account the combined effects of disturbance terms and observer parameters on the width of the convergence region. In the reviewer's opinion, this manuscript is well written and the content of the study is challenging. I hereby give some of my comments and suggestions, which I hope the authors will carefully consider and explain.

1) The reviewer suggests that the authors consolidate the descriptions of the main contributions and main advantages in the abstract and make them more concise to improve readability.

2) How does the proposed algorithm deal with “the combined effect of disturbance terms and observer parameters on the width of the convergence region”? The authors should give reasonable explanations.

3) More explicit comparisons with the existing related literature need to be given to illustrate the novelty of this manuscript.

4) The main challenges encountered in this research work need to be pointed out, and the importance of the proposed algorithm should be further emphasized.

5) In the comparative description of the advantages of the algorithm proposed in this manuscript on page 3, the authors should make a more explicit comparison with the existing literature.

6) What are the practical applications of the proposed observer in engineering, please give some examples.

7) Missing a statement about the environment in the description of the simulation. For example, programming language, computer type, etc. Simulations should start with this information.

8) The caption of the simulation figures is about verbose. The introduction of the parameters and other information in the simulation figures can be placed in the text to make the title more concise.

9) Some background should be enriched, such as neuroadaptive performance guaranteed control for multiagent systems with power integrators and unknown measurement sensitivity, fuzzy-based robust precision consensus tracking for uncertain networked systems with cooperative-antagonistic interactions.

10) The conclusions section is more like a retelling of the abstract, please reorganize the content of this section.

11) The authors are suggested to add some outlook on future work in the conclusions section.

12) There are still some inappropriate expressions in this manuscript, such as some typos and overly long sentences. It is hoped that the authors will check the entire manuscript to avoid these errors.

Author Response

Dear Reviewer 1

We would like to thank you for the time dedicated to review our paper. Next, we attach a PDF file with all the responses.

The authors.

Reviewer 2 Report

This paper presents a new algorithm for setting the parameters of a robust observer for second-order systems, considering persistent but bounded disturbances, along with intended convergence proofs. 

Although the math could be correct, the authors should make an effort to carefully and clearly explain and detail it; there are many incoherences. Constants defined paragraphs before first use, obscure math results (not explicit), careless of formal definitions, et cétera.

Observations that were used to determine my recommendation are:

- "Both Dynamics" in the title says nothing; remove or complete the phrase.

- Please define explicitly states in Equation (1).

- Is the Function (10) continuous, AND a Lyapunov CANDIDATE function? How is differentiated? A Lyapunov candidate must be continuous.

- delta_min and delta_max are never defined; revise that all notations are defined. Since they are not defined, psi_z definition has any sense.

- What an "overall Lyapunov function" is, and what is not?

- The bioreactor model (27)-(29) order is different from the presented for the previous analysis (1)-(2). How the analysis could be extended to that model?

Author Response

Dear Reviewer 2

We would like to thank you for the time dedicated to reviewing our paper. Next, we attach a PDF file with all the responses.

The authors.

Round 2

Reviewer 1 Report

It can be accepted as it is.

Reviewer 2 Report

The authors correctly addressed all of my concerns in this version of the paper; this reviewer recommends the article for publication.